# A New Military Hierarchy of Needs Model

**Márta Pákozdi** [1] **and György Bárdos** [2,*]

1 Doctoral School of Psychology, Faculty of Education and Psychology, ELTE Eötvös Loránd University, Izabella Str. 46, 1064 Budapest, Hungary; marta.pakozdi@hm.gov.hu

2 Institute of Health Promotion and Sport Sciences, Faculty of Education and Psychology, ELTE Eötvös Loránd University, Prielle Kornélia Str. 47, IT Campus F Bldng, 3. Floor, 1117 Budapest, Hungary

* Correspondence: bardos.gyorgy@ppk.elte.hu; Tel.: +36-30-269-8900

**Abstract:** The aim of our research was to create an up-to-date model of the hierarchy of needs of regular soldiers serving in the Hungarian Defense Forces. Our starting point was the theory of motivation, which states that people's actions are motivated by a system of needs. As qualitative systematic data analysis offers an opportunity to create a relevant (decisive) theory to answer the main research question, we based our analysis on the grounded theory method. The results showed that the needs identified in our model often resembled those identified in other theories, including military models, although on different levels. The hierarchy of needs pyramid that we constructed contains terms such as *resources*, *power*, *team spirit*, *development*, *quest for challenges, and fulfillment* and, by way of general background, *appreciation*. Our research broadened our knowledge regarding individuals' motivation to choose a military career.

**Keywords:** soldiers' needs; hierarchy; needs pyramid; grounded theory





## 1. Introduction

### 1.1. Theoretical Background

The first comprehensive model of motivation theory was Maslow's hierarchy of needs (Maslow 1943). According to Maslow, motivation is based on the satisfaction of needs, and it is these needs that dictate an individual's behavior. He also stated that needs can be arranged in a hierarchy. In Maslow's theory, individuals are motivated by seeking to satisfy five categories of need, arranged in order of necessity. These categories are *physiological* needs, *safety* needs, *love and belonging* needs, *esteem* needs, and *self-actualization* needs (Navy 2020).

The lower-order needs (external motivations), such as physiological and safety needs, are intrinsic to human nature. They are prepotent to higher-order needs (inner motivations) such as love, esteem, and self-actualization. Although higher-order needs may be as basic and intrinsic to human nature as lower-order needs, only a small number of people experience them. It is, therefore, challenging for managers to meet the higher-order needs of their employees in practice, since these needs are different in every individual. Motivated employees experience internal tension due to unsatisfied needs, and this tension drives need satisfaction. The stronger the drive, the greater the efforts made to achieve the goal (Robbins and Judge 2009). A good leader recognizes this and reconciles organizational goals to individual motives, thus, making it possible to use tension-reducing behavior to realize the objectives of the organization. According to Taylor and Seager (2021), Maslow's hierarchy is better represented as a dynamic circle that can optimize wellbeing. The COVID-19 pandemic has demonstrated that human wellbeing and health depend equally on biological, psychological, and social factors and that these interact dynamically. The characteristics of hierarchy of needs models were analyzed in detail by Simon (1973). His paper has been cited in many publications as providing a good description of the theoretical background (e.g., Wu 2013).

Interestingly, several authors have pointed out that, although Maslow's theory has been in existence for many years, very few studies have attempted to find supporting evidence (Lieberman and Vrba 1995; Wahba and Bridwell 1976). The hierarchy of needs theory is surprisingly widely accepted, despite the lack of conclusive empirical evidence.

Maslow's theory has, in fact, been criticized on several grounds. The most important criticism refers to the claim that unsatisfied needs motivate while satisfied needs activate a movement to a new level of need (Wahba and Bridwell 1976; Mercado 2018). Another criticism is that the spectrum of human needs is far wider than that presented in Maslow's system, and employees are unable to differentiate among Maslowian needs or recognize the minute differences between them (Vargas-Hernandez and Arreola-Enriquez 2017). A further criticism refers to the huge individual differences in higher-order needs and to the fact that self-actualization means different things to different people (Sosteric and Ratkovic 2020). While some people regard their daily successes as self-actualization, for others, self-actualization means achieving life goals through perseverance and hard work.

Since Maslow's initial theory, many other theories of needs have been formulated. Due to the ubiquity of the original model, several authors considered it necessary to compare their preferred models with Maslow's. Taormina and Gao (2013) conducted a study to measure the satisfaction of needs (n = 386) and found that predictors such as family support, traditional values, and life satisfaction had significant positive correlations with the satisfaction of needs. On the other hand, anxiety and worry were negatively correlated with the satisfaction of needs. Multiple regression analyses support Maslow's theory that the satisfaction of lower-order needs predicts the satisfaction of higher-order needs.

Gawel compared Maslow's theory to that used in business literature for the analysis of needs to identify which of them better describes business needs (Gawel 1996). Gawel found that the needs of participants in the Tennessee Career Ladder Program (TCLP), especially teachers, motivated them to behave differently from business employees and that neither Herzberg's nor Maslow's theory perfectly described this difference in behavior. Maslow's theory was also compared to Herzberg's two-factor theory (Mausner and Snyderman 1993) in a comparative study (Velmurugan and Sankar 2017), which concluded that Maslow's theory generated more important values in many organizations compared to the two-factor theory.

Rauschenberger and his colleagues compared Maslow's theory to that of Alderfer (Rauschenberger et al. 1980), using a Markow chain for the comparison. They concluded that both theories were disconfirmed and even that the concept of the hierarchy of needs as a whole should be avoided. The question has also arisen as to whether Maslow's theory can be used to test quality of life (QoL), especially in geriatric patients (Chang and Hsiao 2006). Although, in this population, the hierarchy of needs model and the applied QoL questionnaire did not show complete agreement, the hierarchy of needs approach seemed to be useful and applicable at a general level.

A valuable study was conducted on the use of Maslow's concept in social media adoption (Ghatak and Singh 2019). The hierarchy of needs model was shown to be a useful basis for a social media adoption model, while a scale developed on this basis could be a valuable tool for the launching of various marketing campaigns.

An interesting model was created by Moorhead and Griffin (2008) in the form of a motivation circle (see Figure 1), in which needs, and the behavior required to satisfy those needs form an endless circle. Independent of the respective theory, there is shown to be a universal and basic form of connection between motivation and behavior.

In summary, significant ambiguity exists with respect to hierarchy of needs models. While some authors are strongly critical of Maslow's hierarchy of needs, others still consider it to be the best model. Having reviewed the complex literature, we concluded that no individual theory can be regarded as superior, but Maslow's theory can be used as a reference.

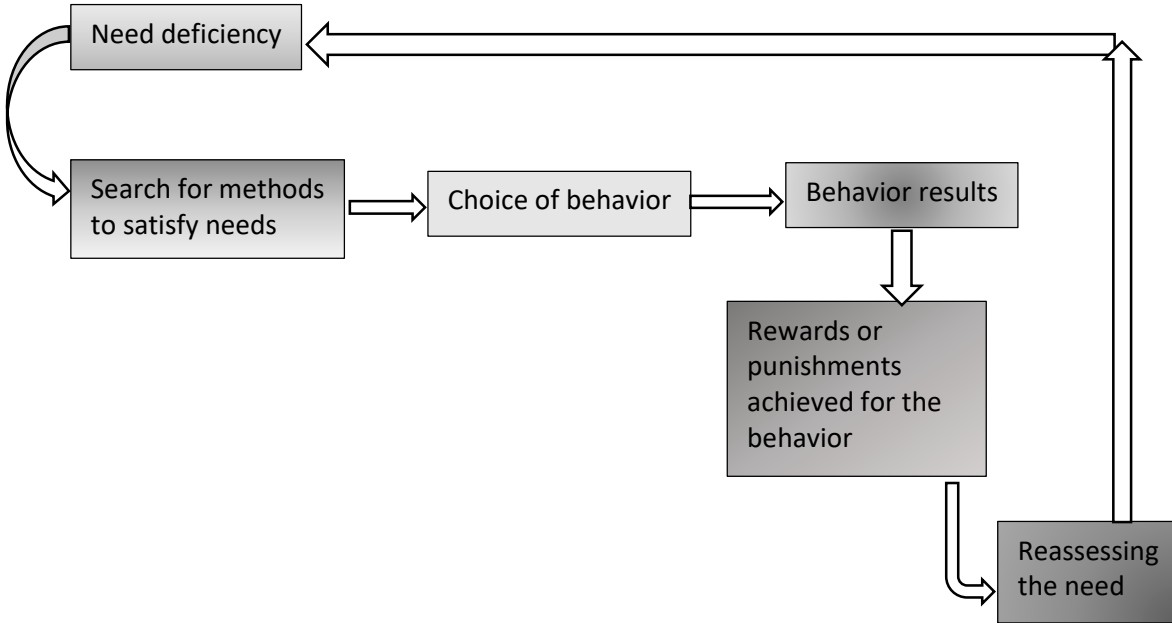

**Figure 1.** Motivation is a circle, a process without an end (based on: Griffin and Moorhead 2014, p. 91).

*1.2. The Military Hierarchy and Its Components*

Military organizations and structures can be seen as representing a special category among the available models. One such model deals with the relationship between the demands of the military and the potential resulting mental health problems (Adler and Castro 2013). These problems are associated with both organizational and individual aspects and affect how members of the military deal with military service. The model suggests that soldiers' capabilities should be assessed before they become involved in potentially traumatic projects. It also suggests how mental health problems might be managed in the military.

One important question concerns how different coping strategies affect the resilience, psychological wellbeing, and perceived health of military personnel. One conceptual model indicates that approach-oriented coping strategies improve resilience, which, in turn, directly affects psychological wellbeing and perceived health status (Chen et al. 2018).

Several models deal with human resource planning and the motivational background of recruits. An experimental model was tested in the U.S. Army Reserve to identify the major factors in human resource planning needs (Reeves and Reid 1999). In another study, Mosko's institutional–occupational model was compared to public service motivation (PSM) studies, and a positive correlation was found. This suggests that public management researchers should usefully pay greater attention to scholars interested in military recruitment, while Mosko's model offers a sound background for this process (Taylor et al. 2015). Another study (Pliakos-Trypas 2019) explored what motivates military personnel in Greece to perform better and determined the fundamental elements on which motivation should rest in the Greek army sector. The research focused on the cultural characteristics (norms and values) of the Greek military environment.

A review of major surveys undertaken on motivations and reasons for enlisting in the U.S. military was published in 1983. Besides general attitudes, the unemployment rate, military pay, parental occupation, and education level, mutually interactive factors were also identified as important (Boesel and Richards 1983). In 1986, a scale was developed by the United States Army Research Institute to characterize the enlistment motivation of new recruits. The New Recruits Survey (NRS) features two subscales, one for the Active Army and one for the Reserves. The two subscales share some common components, while other components are different. This research provided the basis for improved testing and

emphasized the importance of such measures (Baker 1990). Enlistment motivations in the U.S. were also studied in 1988, when young adults were asked about their motivations for joining the military. The Career Decision Survey demonstrated that it was mainly psychological factors that played a role in the decision-making process—that is, thoughtful decision making. However, in addition to these rational factors, a generalized emotional response was shown to have a significant impact on this process (Wilson and Perry 1988). Military operational contexts require military personnel to develop technical and nontechnical skills for the performance of tasks and missions. Nontechnical skills predominate in military teams. The results of the survey indicated that situational awareness, decision making, communication, teamwork, and team leadership are the most important values for military teams. A hierarchical skills development scheme was proposed for nontechnical skills that are fundamental to the military context (Cavaleiro et al. 2022).

The motivation of military personnel has been shown to play an important role in whether recruits stay in or leave military service (Ibrahim et al. 2021). However, it has also been emphasized that researchers and practitioners frequently fail to recognize that the model they either reject or support is an inaccurate representation of actual military command and control in both doctrine and practice (Burke 2018). The central aim was to eliminate inaccurate assumptions regarding the command and control model while persuading critics to adopt a new and more informed assessment of the military model in its modern form.

The role and situation of women in military service represent a special topic. The wellbeing of women in the military is affected by many different factors, such as career, family life, individual characteristics, military events and obligations, and resources (Segal and Lane 2016). In the context of a qualitative (grounded theory) analysis, semi-structured interviews were conducted with 18 enlisted female service personnel, who were asked about opportunities, calling, and outcomes. In the case of half the interviewees, enlistment had resulted in a professional military career. The study highlighted the need for research dealing with the motivation, retention, and reintegration of women in a progressively more hostile work environment (Mankowski et al. 2015).

Special circumstances (such as the COVID-19 pandemic) have also been studied. The COVID-19 pandemic is regarded as a "threat multiplier" throughout the world—that is, as a significant and new type of assault affecting the way in which society operates. It poses an additional threat because "traditional", existing conflicts have not disappeared but remain present worldwide.

However, the fact that military organizations have previously faced similar problems, such as an influenza pandemic and other diseases, is reassuring in that several methods have already been developed to ensure the operational capacity of the military. This is a promising factor in the pandemic era, as it suggests various ways in which the situation can be improved (Scott et al. 2020). The Hungarian Defense Forces have also introduced new solutions that may contribute to the survival of society. This is, essentially, the topic of the present article.

As described by Wu, there are both nested and non-nested hierarchies. This is relevant to the present topic, since armies consisting of soldiers of different ranks are typical examples of nested hierarchies (Wu 2013). A survey carried out by NATO between 2010 and 2013 in 22 member states, including Hungary, was targeted at personnel serving in the military medical service. It identified a need to place greater emphasis on the military's ability to retain personnel; it is not enough to recruit new personnel, since there is a need for well-trained soldiers. The NATO survey found that the factors that contribute to the retention of military medical personnel are the following, in decreasing order of importance: tours of duty abroad, higher salary, better working conditions, and increasing/decreasing patient relationships (Report Responding to the COMEDS 2015).

The U.S. Army is concerned with both the physical and mental wellbeing of its soldiers. Based on the hierarchy of needs in the Maslow pyramid, it conducted a survey in 2017 on the basic needs of 4856 non-commissioned officers (NCOs). Security and food appeared to

be the dominant needs among the respondents (ADP 6–22 2019). In his model, Sirgy (1984) correlated Maslow's hierarchy of individual needs with the needs of society. He concluded that societal needs (the physical needs of citizens, sovereignty, global commerce, honor) are similar to individual needs and must be met to improve quality of life.

*1.3. Civil–Military Interaction*

To obtain a better understanding of motivational needs and the special factors that affect subjective feelings towards military service, the characteristics of civil–military interaction must be explored. The military and society are related in various ways, which can sometimes change significantly. This fluctuating relationship is influenced by several factors in both civilian and military life which can sometimes change very rapidly and profoundly. Changes in society can even be shown to affect enlistment in the military—for example, the ratio of women to men among new recruits (Shields 2020). The relationship between society and the military may differ significantly in different eras. The two may become closer or more distant depending on the local and global political situation. This doubtless means different things for long-term members of the military and for those who are contemplating, or on the point of entering, military service (Shields 2006).

The relationship between the military and civilian society depends on several factors, including, for example, the attitude of politicians and their willingness to include or invite members of the military into the political field, the actual structure and leadership of the military forces, and the tasks and operations of the military. This explains why the relationship is different in different countries and why detailed studies are needed before coming up with an overall view or launching a common operation (Harig et al. 2021). Civil–military interaction has been a focus of recent research not only in the U.S. but also in many other countries. Although countries differ in many respects, the relationship between civilian society and the military shows surprising similarities among them (Egnell et al. 2014; Gjørv 2016). Some interesting approaches to characterizing the background to this special relationship can be observed. Based on the pre-trained language model, a Chinese military relation extraction method was presented (Lu et al. 2021) that combines the bi-directional gate recurrent unit (BiGRU) and multi-head attention mechanism (MHATT). Specifically, the conceptual foundation for this method lies in the construction of an embedding layer and the combination of word embedding with position embedding, based on the pre-trained language model.

Society has preconceived notions of soldiers as being brave, strong, heroic, steadfast, respectful, humble, committed to their homeland, and ready to sacrifice their lives for their fellow human beings and for their homeland. However, soldiers are also human beings and are affected by external events, such as the present pandemic, while the public expects them to handle their emotions effectively so that they are able to ensure security and protect people's lives (Pákozdi and Fejes 2015). Observation of soldiers' behavior over several years of service, as well as studies conducted with soldiers at work, led us to the decision to create *a hierarchy of needs model for soldiers.* Our studies focused on soldiers' satisfaction with military service. The findings of our earlier psychometric study (n = 120) among personnel in the military medical service showed that an increase in salary did not, in itself, motivate them to remain in the military. If satisfied, esteem, which is a higher-order need in Maslow's pyramid, serves as a stronger motivation to continue in a military career (Pákozdi and Fejes 2015). On the other hand, the results of another qualitative study among active service personnel (n = 288) showed that respondents anticipated a positive future for the Hungarian Defense Forces and that they expected an improvement in living and working conditions, which implied dissatisfaction with the way their basic needs were met (Pákozdi and Bárdos 2022).

To improve the military's capabilities to retain personnel, it is essential to know what it is like to be a soldier at present in this country. What factors motivate a person to stay in the military and regard military service as a profession? How do soldiers' motivation profiles change during their time in service?

*1.4. The Research Agenda*

The aim of the present study was to create an empirically based model of the hierarchy of needs of soldiers in the Hungarian Army, a typically East-Central European entity with the specific background of this region. Despite sharing common features, armies in different countries or regions have different characteristics in terms of the relationship between the armed forces and society. A further aim of our research was to demonstrate that the hierarchy of soldiers' needs can be used and taught both in psychology and military science.

The study was characterized by two special features—the methodology and the country—since we applied a qualitative approach and methodology which are seldom used in this field. We believe that the qualitative approach of this model provides a useful tool that will help researchers and lecturers to understand the examined phenomena (Taylor et al. 2015). The method used here may have helped to obtain additional information that would not have come to light in a traditional, questionnaire-based study. The present article summarizes the findings of a three-year research project based on a qualitative design. Our hypothesis was that the analysis of the interview transcripts would allow us to learn more about the basic features of the hierarchy of needs among Hungarian soldiers and would provide an opportunity to obtain new insights to add to our existing knowledge in this field.

Qualitative data analysis can be seen as suitable for the formulation of new views and correlations and for the reorganization of previous knowledge and the content of existing concepts (Somogyi et al. 2018). Systematic data analysis can provide opportunities to create a relevant (decisive) theory to answer the main research question (Corbin and Strauss 2008). To date, as far as we know, no questionnaire has been able to reveal the inner feelings and motivations of soldiers, which is why we decided to use a qualitative method that is better suited to this purpose. The main feature of this method is that individuals are free to talk about their inner feelings without the external pressure or directive of preset questions.

Military sociology is a useful framework for the study of civil–military relationships and of the background to the transition from civilian life to military life and vice versa. There are a significant number of publications dealing with this complex topic which describe how the military is organized. Although these relationships differ according to both geographical location and period, their results are important for anyone intending to analyze society and the military as two separated but connected identities (Shields 2013). The qualitative approach has been successfully applied in military research as well and has found its place in the methodological pool. One comprehensive textbook of military research methods (Soeters et al. 2014) describes several approaches, including a representative overview of qualitative analysis (Rietjens 2014). Other publications reinforce this view, describing how the qualitative approach can be applied in military research (Bernard et al. 2010; Flick 2009).

The key outcome—the development of the motivational profile of a military career—led to the compilation of a multifactor motivational questionnaire, which may help to determine possible interventional approaches. The validation of this questionnaire is in progress and is due to be published in a forthcoming article.

## 2. Materials and Methods

*Sample*

The respondents were selected from among personnel who had been in a legal relationship with the Hungarian Defense Forces for at least five years, which was a precondition for participation in the research. This inclusion criterion was necessary because we needed respondents who had already been socialized into the military, were familiar with the system, had a personal involvement, and had made an individual decision, all of which have a significant impact on a person's motivations. In the first two years (2018 and 2019), data were collected during semi-structured group interviews. However, the data collection method had to be modified due to the COVID-19 pandemic in the third year (2020). Since the lead author is both a researcher and a military officer, sample selection was straightforward. Participation in the study was voluntary and anonymous.

The group interviews were conducted during internal courses, at work meetings, or at special meetings dedicated to the research. An invitation to participate in the study was sent in advance to these locations. Our aim was to cover the whole country, although participation was voluntary since the research was planned as an exploratory rather than a representative study. We wanted to obtain personal and truthful opinions; thus, we did not collect data on gender, age, rank, time in service, or military branch, which might have distorted the results. However, our intention was to interview equal numbers of officers, non-commissioned officers, and regular soldiers, as well as equal numbers of men and women, despite the fact that there are no "male soldiers" and "female soldiers" in the military forces, merely "soldiers."

The interviews took place in groups of a maximum of 10 people with the seats arranged in a semicircle and lasted for 2 h (Fassinger 2005; Ponterotto 2005, 2010). As moderators (O'Donell 1988), during the interviews, we ignored the military regulations regarding communication with superiors, which greatly contributed to the expression of honest opinions (Krueger 1988, 1993; Corbin and Strauss 2008). To obtain information about the participants' motivations, their attitudes towards the armed forces, and their ways of thinking (Barbour and Kitzinger 1999), the respondents were asked to explain why it was good to be soldiers and why they liked living as soldiers ("Why do you like being a soldier?"). As this was the only question, participants had sufficient time to express their opinions in turn. The moderator asked them to focus on this topic and ignore other (though doubtless important) areas. During the interviews, where it was considered necessary, we asked adherers/deepeners ("Why do you think it is good to go on missions?"), clarifiers ("Do you mean you would like to stay in the army your whole life?"), or detailed questions ("What do you think about the relationship between family life and the military profession?"). When answering these questions, the interviewees elaborated on their opinions, emotions, and thoughts.

Following the outbreak of the COVID-19 pandemic, rather than group interviews the participants were asked to write narratives. At this time, we also focused on how the pandemic was affecting their military careers, although this topic is omitted from the present paper. The study was conducted in several rounds in the framework of forums (e.g., workgroup meetings, military events, etc.) that involved representatives of different military organizations. Participants were selected using the random sampling method (n = 28) and spent between 10 and 15 min writing their narratives. We were not present while the participants were writing; we simply greeted them at the location at which they had been asked to participate in the study. The narratives were collected in labelled boxes at each location. After reading the 42 narratives collected in the study, 14 were excluded because the respective participants did not fulfil the minimum requirement of five years in service, which was the precondition for inclusion in the sample. As a result, narratives written by 28 participants were analyzed. The average length of military service among the participants was 15.54 years (SD: 5.203, min. 5 years, max. 24 years).

When writing the narratives, participants were requested to complete the sentence "*I like being a soldier because . . . .*", since it was important for the narratives to be based on the respondents' own experiences and perceptions. Transcripts of 49 group interviews (a total of 386 participants), as well as the written narratives (n = 28), were then analyzed. We compared the answers given during the interviews with those obtained in written form and found no significant differences. The two sets of responses were, therefore, analyzed together.

## 3. Analysis and Coding

In this type of study (i.e., structured interviews), questionnaires that contain preset questions are typically used. However, this restricts the information obtained and prevents the expression of different opinions, even if the questionnaire is created to obtain data on specific topics (e.g., see Filosa et al. 2021). By comparison, using a qualitative methodology yields wide-ranging data, including data that are completely new and original. We, there-

fore, chose a qualitative approach to obtain more information about the inner thoughts and opinions of the soldiers.

The qualitative analysis was carried out using the grounded theory narrative method (Bernard et al. 2010; Bryant 2017; Chamberlain 1999; Charmaz 2008; Khan 2014; Corbin and Strauss 2008; Pákozdi and Bárdos 2022; Strauss and Corbin 1997, 1998). The interviews were recorded, then, transcripts were made so that we had a written version of the interviews.

During the analysis, three levels of coding were applied: open (basic) coding, axial coding, and selective coding. This process resulted in fewer and fewer codes representing larger and larger categories, leading to a final, concentrated, complex overview of the interviews. At each level, the elements of the previous level were compared, collected into groups of similar characteristics, and named according to their collective meaning. This process resulted in ever smaller numbers of increasingly complex categories.

The applied grounded theory method does not involve statistical analysis and quantification; thus, it provided for more intensive data collection. The analysis is a bottom-up process in which the text is elevated to new conceptual levels in several steps, generating a hierarchical structure that assists theory formation (Charmaz 2008; Glaser 2014). In this way, a systematic, inductive, interactive, and comparative model could be established (Wertz 2011). Grounded theory is an analytical method that initially pushes to one side (ignores) the researcher's theoretical knowledge (referred to as the state of theoretical agnosticism by Charmaz 2008) and emphasizes the knowledge and information found in the text. The identification of the essential factors is based on the triple coding process characteristic of the grounded theory method (basic/open, axial, and selective codes), which helps to identify, open up, and interpret the essence of the findings (Charmaz 2008).

Figure 2 shows an example of the coding process, during which the original texts were individually scanned for characteristic components, sentences, or word combinations, which were then assembled into a common collection. Within this collection, similar expressions were grouped, while those that had no relevance to our topic were eliminated. The groups were then compared to the original texts to ensure that the interpretation was correct and corresponded to the thoughts expressed by the soldiers.

**BASIC (OPEN) CODES**

authority, behavior, belief, compliance, courage – endurance, difficult carreer, done well, family, family identity, gives a frame, hard work, health, honor, housing allowance, humility, life career, non-competitive salary, power, pride, profession, professionalism, purpose, secure livelihood, secure position, sport, training, travelling, uncertainty, uniform, viability.

**AXIAL CODES**

| Military Track | Organizational Culture | Looking for Safety |
|---|---|---|
| family identity | humility | secure position |
| training | viability | uncertainty |
| difficult pathway | belief | housing allowance |
| travelling | pride | non-competitive salary |
| hard work | compliance | secure livelihood |
| done well | behavior | |
| life career | professionalism | |
| uniform | gives a frame | |
| profession | courage – endurance | |
| family | honor | |
| power | purpose | |
| authority | health | |
| | sport | |

**SELECTIVE CODE**
**RESOURCES**
**military track**
**organizational culture**
**looking for safety**

**Figure 2.** An example of the analytic process—basic codes (first step) to axial codes (second step) to selective code (third step).

### 3.1. Open Coding

After reading through the transcripts several times, we first created open codes by assigning categories to the meaningful segments of the texts (i.e., words, expressions, or sentences). In this phase, we examined the context of the codes—that is, their antecedents, circumstances, and consequences, their environment, and their space and time frame. In the course of further processing, some categories were retained while others (e.g., those containing similar elements or non-relevant phrases) were rejected. The basic idea was to find out what a specific category can tell us about the basic problems, components, and aspects of the studied phenomenon. In the above example, the initial analysis resulted in 30 basic codes.

### 3.2. Axial Coding

During the axial coding process, our aim was to identify conceptual relationships among the open codes by investigating what a specific category can tell us about the basic problems and components and the different aspects of the studied phenomenon and by looking for associations between individual codes and different dimensions. By the end of the axial coding process, basic categories and subcategories were identified, allowing us to find relationships and correspondences. The notes taken in this phase contributed to the analysis in the next phase of the work. In the example shown in Figure 2, the 30 basic codes are arranged into three groups under the representative axial codes. The three axial codes contain 12, 13, and 5 basic codes, respectively; thus, the number of basic codes collected into one axial code is not predetermined.

### 3.3. Selective Coding

During the final, selective coding process, the categories and subcategories obtained in the first two phases were collected and grouped under new, higher-level, summarizing codes and compared against the transcript (i.e., the correlations between the codes and the transcript were checked so that the correspondences could be validated). In this phase, we did not use the categories that were omitted during the axial coding process. In the example, one selective code contains the three axial and the 30 basic codes (the so-called code tree). It should be noted that there are selective codes to which no axial codes were added; instead, the open (basic) codes were included directly (see Figure 3).

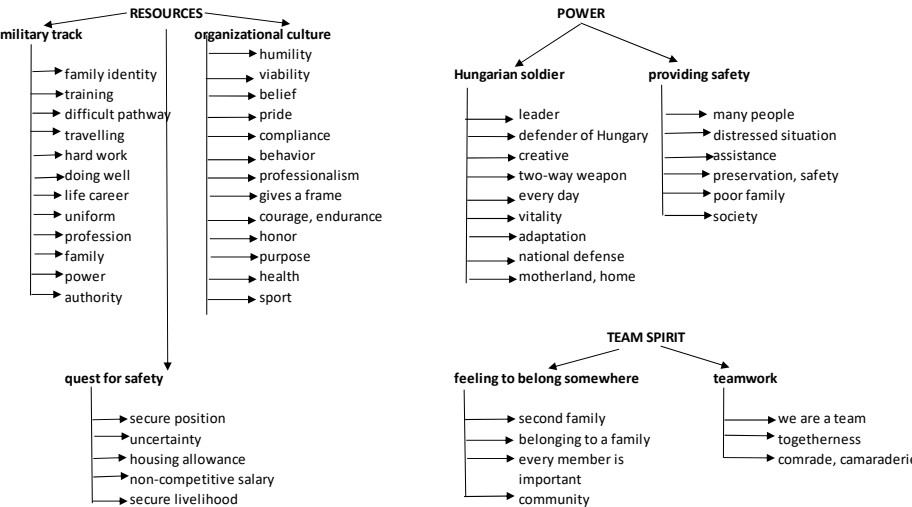

**Figure 3.** *Cont.*

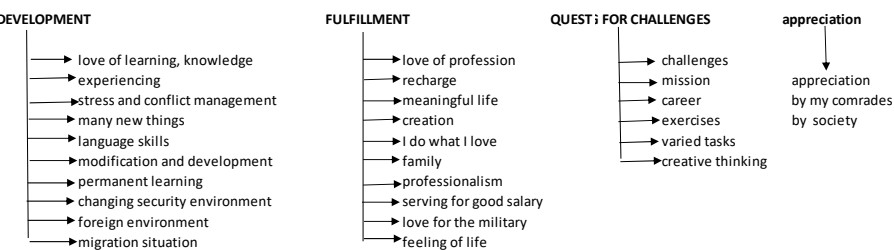

**Figure 3.** Selective codes and the terms belonging to them.

Although separate selective codes were created during the coding, they were not, of course, entirely independent from one another. This is represented in Figure 4, which shows the selective codes and the subcodes through which they are connected. This helped give the researchers a general and comprehensive impression of the deep structure of the sample.

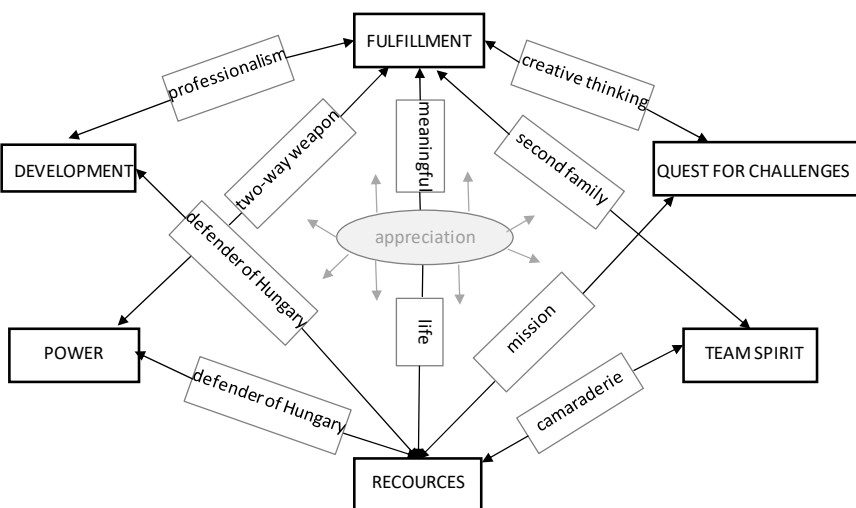

**Figure 4.** Overview of the selective code network.

Based on the code trees, we first created a star-shaped model of the selective codes and, ultimately, the military motivation pyramid 2021, which represents the new hierarchy of needs structure identified in the research (see Section 5 below).

## 4. Results

*Codes*

The texts referred to factors that affect whether a person stays on the military pathway, while the soldiers described their feelings in terms of external intention. The narratives were generally written in the first person singular and in the present tense, although social experiences were described in the plural. The narratives sometimes mentioned negative factors, expressing possible dissatisfaction with the military profession. All the texts were self-narratives written by soldiers in the Hungarian Defense Forces.

RESOURCES, which is the first selective code, is associated with being a soldier and formed the central element of the narratives in a unique and peculiar way. All the self-narratives dealt with the diversity of the **military track** (i.e., career), from *family identity* to *career*, that can be achieved through self-belief. The diversity of the military career is indicated by military *training*, during which the individual's military socialization begins. In addition, a military career is also a *difficult* career, which requires *hard work* and *power* and which must be *done well*. This basic category also reflects the *professionalism* associated with a *life career*, which is a possible main motivation for joining the army. The *family* is a reference entity, which also represents an *identity* as a basic RESOURCE. Thus, profession

and family are both important to the soldiers, and there is no way to choose between them. The only way is to find a balance. RESOURCES also encompass *authority*, since the soldier reflects authority itself and, as weapons of authority, soldiers defend their homeland even by the sacrifice of human lives, in which patriotism is the dominant factor. This is also associated with the *uniform*, a symbol that mediates values, such as belief and camaraderie, which are elements of everyday military service.

Besides the soldier's self-belief, RESOURCES is reinforced by elements of the **organizational culture**, such as learned and acquired (via missions and trainings) knowledge, patriotism, and homeland protection. In addition to values such as *courage* and *professionalism*, there is also an emphasis on *behavior* and *compliance* as providing a *framework* for the soldier as an individual. The organizational culture also *gives a framework* for soldiers entering the service, who require basic features such as *humility*, *viability*, *belief,* and *pride*, as well as *courage*, *endurance*, *honor*, and *purpose*, which characterize attitudes toward military service. *Health* and *sports* are also expected values and norms among the soldiers that appeared in the self-narratives as expectations towards the military.

Safety, as a basic factor in itself, also comes under RESOURCES in the form of the **quest for safety**. Demands, such as *secure position*, *secure livelihood*, and *housing allowance*, to counteract and balance future *uncertainty* and a *non-competitive salary* are also central elements of the narratives.

The second selective code is POWER, which represents the essence of military service and the capacity of the soldiers to meet its demands. Participants endeavored to characterize the **Hungarian soldier** as meeting demands and fulfilling future assignments in the army. A Hungarian soldier must be a *defender of Hungary,* whose task is to defend the *motherland* and the *home every day*. The term *two-way weapon* belongs to both the peacetime and wartime (armed conflict) dimensions and refers to peace building and peacekeeping—that is, security. The experienced participants characterized the ideal soldier in the Hungarian Army as someone who is *creative*, able to *adapt* to different situations, committed to *national defense*, serves with *vitality*, and, sooner or later, can become a *leader* of their group. High self-esteem and the pursuit of excellence during task execution are also associated with one aspect of POWER—that is, the fact that the soldier is able to exercise self-control and experience self-power every day. This aspect of service generates the feeling of power; resilient, well-adapted soldiers consider the defense of their homeland as their primary mission, which is clearly distinct from dominance and authority.

Patriotism is predominantly reflected by the expression "serving my homeland", which featured in all the narratives. Behind the term *homeland*, *home*, and *Hungarian Army*, we can identify a high level of commitment. Military assistance reflects liking towards members of society, which is associated with **providing security**—that is, security for *many people,* even for the whole of *society*. This is especially true for those in *distressed situations* and for *poor families* (even frequently for their own families) that require *assistance* to *preserve safety*.

Besides the uniform as a symbol, TEAM SPIRIT has a strong impact on both soldiers (as individuals) and society. The **feeling of belonging somewhere**, as a subcode, was an active element in the narratives. An internal demand among committed individuals is apparently the desire for a *second family*, *belonging to a family*, and *community*, in which *every member is important*.

The phrase "*We are a team*", as an element of **teamwork** (a definite subcode), as well as *togetherness* in achieving goals, characterizes both the organization and the individual soldiers. *Camaraderie* and *comrade* are basic terms applied to the troops, suggesting that the soldiers know and feel that they can count on their fellows, who would do what any soldier would for another. This human value was a recurrent element in the narratives.

DEVELOPMENT requires adapting to a *changing security environment*, sometimes even to a *foreign environment*, when on missions or as a result of the *migration situation* by developing *language skills* and learning how military technology operates. Handling *intense stress* and/or *conflicts* also requires adaptation and development. This extreme need was mentioned in many of the narratives.

New assignments generate *experience* and *knowledge* obtained by means of *permanent learning* that results in the understanding of *many new things* not only on the battlefield but in peacetime too. A *love of learning* is the basis for change and development and is a need that was referred to in almost all the narratives.

Alongside *missions* as the major challenge facing the military profession, QUEST FOR CHALLENGES also encompasses training *exercises*, participation in which is also dependent on posting. These two terms are also connected with the need for rapid advancement in one's trajectory and *career*. One characteristic expression is "I have *tried* myself in a mission in a hot situation", which helps the individual to recognize their limits. The need for *varied tasks* also belongs under QUEST FOR CHALLENGES, along with *challenges* and *creative thinking*, which is associated with the effective combination of learned and experienced knowledge.

M*eaningful life*, as a military feeling, representing FULFILLMENT, makes it possible for soldiers to do what they like doing, allowing them to become *recharged* and, thus, improve their *professionalism*. *Family*, as well as being present as a feeling of belonging somewhere, is mentioned in the narratives as the soldier's own family, suggesting that a high level of self-knowledge and need drive the quest for happiness and self-realization that can be achieved even in the Hungarian Army. Elements of FULFILLMENT, such as *serving for a good salary*, *love of the profession* ("*I can do what I love doing*"), providing safety for the *family*, and *love for the military* profession, were true values among the soldiers and factors that contribute to their satisfaction. A *feeling of life* frequently results in the *creation* of new thoughts and solutions and the fulfillment of the requirements of military service.

**Appreciation**, as an independent axial code, was a basic need among all soldiers that featured in every narrative. It is so strong in itself that it appears as an independent need (i.e., a separate axial code) connected to virtually all the other codes. The sentence "I am *appreciated* by my *comrades* and, I hope, by *society*, too" best symbolizes this desire for recognition. The term *valuable soldier* suggests the virtue of providing safety for fellow human beings, which contributes to the soldier's appearance as something of value within society and to their prestige.

## 5. Discussion

Figure 5, "The star-shape model", demonstrates how needs originate from RESOURCES and return there too.

TEAM SPIRIT, which is also associated with RESOURCES, is reinforced by the uniform. For soldiers to be able to complete their missions, they must believe in themselves and know that they are able to do their job. RESOURCES provide the basis that allows soldiers to develop and thrive, making it possible for them to defend their home country and fellow citizens, including their comrades. The socialization of soldiers becomes complete through continuous DEVELOPMENT. The QUEST FOR CHALLENGES is present when a mission results in an "I'm ready" state for them. Based on everything they have learnt and built together with their family, FULFILLMENT, at the top of the network, generates the feeling of the military life. This requires belief, hard work, and determination, which are available to all.

**Safety**, which is an element of both RESOURCES and POWER, is present in the form of two subcodes in the sample. The **quest for safety** encompasses *secure livelihood* and *secure position*, which are basic needs for everybody, while **providing safety** contains *assistance* to *many people* in *distressed situations*. Since safety is a basic condition, there was no need to show it as a selective code; we merely included it in the model for the sake of completeness.

The connection between TEAM SPIRIT and the provision of safety can be found in the *second family* that emerges through the strength of the community in which *every member is important*, although this aspect may require further study. At the same time, the relationship between patriotism and safety is based on valid experiences, since the main task of the army is to ensure security for Hungarian citizens.

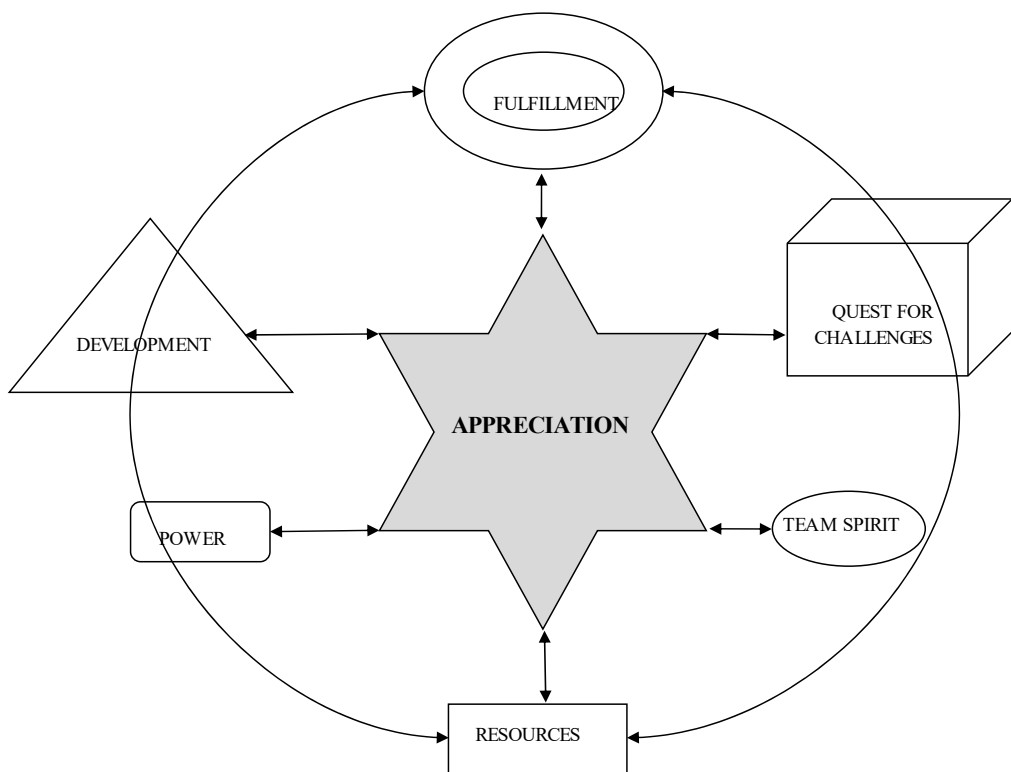

**Figure 5.** The star-shaped model of the selective codes showing how appreciation is related to all of them.

APPRECIATION, which points in every direction in the network illustrated in the star-shaped model, helps soldiers to do their job effectively. Without it, even if they do what they have to do, they will have no positive feelings, which may affect the whole organization neutrally or even negatively.

An appreciated soldier has a value; learns and evolves; provides security; is a good comrade; and has a positive feeling about life. We found no relationship between **Appreciation** and QUEST FOR CHALLENGES in this sample, although further research may reveal hidden connections.

Compared to the military needs pyramid in the theoretical model, the results of the present research differ in terms of both Maslow's hierarchy of needs and the phrasing used (Figure 6).

At the base of the pyramid is RESOURCES, with POWER and TEAM SPIRIT above, representing the basic needs of a soldier. RESOURCES and POWER comprise military socialization, organizational culture, the family, and the belief of the soldier, representing the total resources that, together with TEAM SPIRIT, can help soldiers to overcome difficulties. Although camaraderie, as a value, does not meet the criterion for character strength as stated by Seligman (Peterson and Seligman 2004), it is a factor that, in addition to emotional support, contributes to the soldier's physical stamina.

TEAM SPIRIT, here represented as a basic need, is associated with belonging somewhere and teamwork and corresponds to the liking/belonging aspect of the higher-order needs.

The next element is DEVELOPMENT, meaning learning processes and hard work, which was included in some form in all the narratives. In this model, the term is included as an individual need towards the peak of the pyramid, which, like QUEST FOR CHALLENGES, is a typical feature of the present-day political and military situation.

FULFILLMENT, engendered by the military feeling of life, is also connected to self-actualization and appears here at the peak of the pyramid. It encompasses needs and strengths that contribute to professionalism and that are, at the same time, drivers of a happy life (Lyubomirsky 2009), providing a meaningful life for the soldiers.

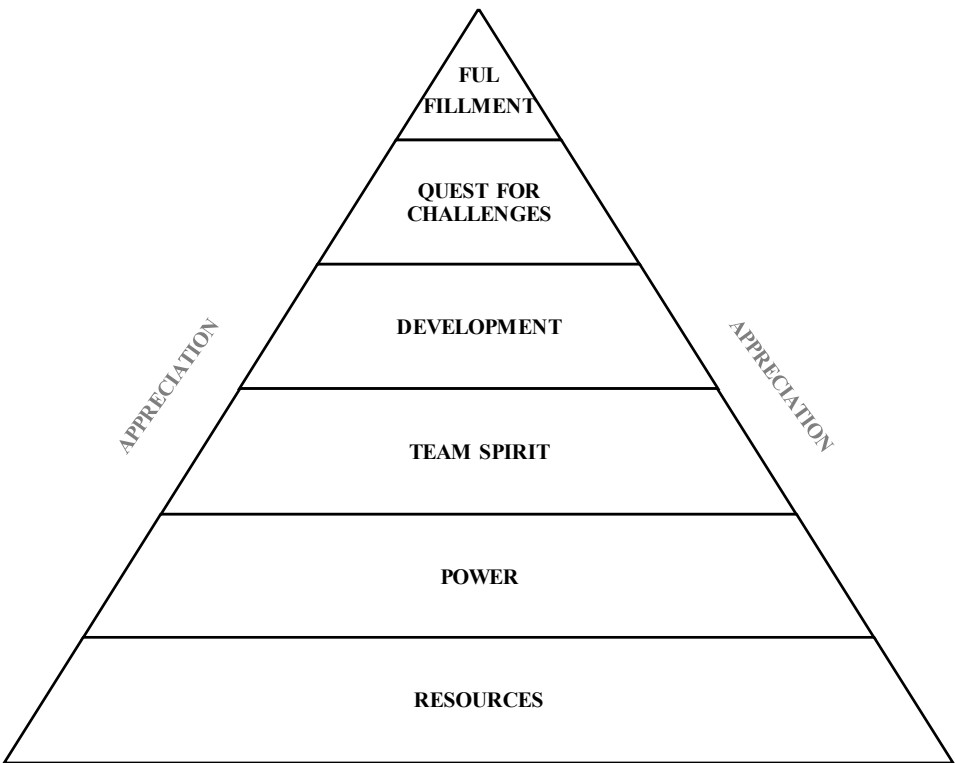

**Figure 6.** Our military motivation pyramid, 2021.

APPRECIATION permeates the whole pyramid; it is so strong a need that it is present at all levels simultaneously. It appears to be a general factor and expectation throughout the need network and a basic motivation for anyone considering joining the Hungarian Army. This is another topic that deserves further research.

In summary, the results show that the needs identified in our model are, in many cases, similar to those in other theories, although at different levels. This suggests that it is worth looking for additional relationships between Maslow's self-realization and the higher-order and basic needs in this model.

Our results are essentially in harmony with the findings of other studies on enlistment motivation among military recruits and the motivations and psychological state of those serving in the military. According to one study, the motivational factors for enlisting in the all-volunteer force varied greatly and tended to be positive and favorable, although some negative, less favorable factors were also identified (Johnson and Kaplan 1991). The factors affecting motivation appear similar to those found in the present research. According to Griffith (2009), there may be many different background factors behind joining the reserve military service. While social identity contributes significantly to joining, personal identity does not. The former contributes significantly to commitment to the unit and to the perceived readiness of the unit and fellow soldiers, while personal identity is positively related to individually relevant outcomes, reporting for duty for contractual reasons, and personal readiness for combat. These findings may help to improve the enlistment process and the readiness of the applicants and can be applied in most countries. In addition to regional characteristics and family traditions, other factors may play a significant role in terms of motivation to enlist in the military. Two factors emerged that play an important role with respect to enlisting: personality and genetic predisposition. This explains why heritability is frequently discussed with respect to military service, which is probably, at least in part, a function of personality traits (Miles and Haider-Markel 2019).

Members of society have different views regarding why citizens join the military. There are two main approaches. One group, mainly comprising liberals, believes that the main motivation is economic—that is, it provides a better salary than other occupations.

The other group, mainly comprising conservative Americans, has an idealized image of applicants as being motivated by self-sacrificing patriotism. Another motive attributed to applicants is the desire to escape from desperate circumstances (Krebs and Ralston 2020).

Similar studies have been conducted in countries other than the U.S., which help to reveal similarities and differences (e.g., Pákozdi and Bárdos 2022; Ibrahim et al. 2021; Pliakos-Trypas 2019). One interesting factor with respect to enlistment motivation is the ability to convert military resources to social resources or objective rewards in the civilian sphere and labor market. Soldiers may be motivated by the possibility of acquiring capital during their military service that can be converted in their civilian life. This is characteristic of countries such as Israel and Great Britain (Grosswirth Kachtan and Binks 2021). The changing international situation, as well as internal problems, has prompted both the Swedish and Norwegian armies to rebuild and enlist new soldiers. A qualitative thematic analysis found at least three major motivational factors: the military as a steppingstone; international missions and geographical location; and benefits (Österberg et al. 2020). These results show that motivations for enlisting and factors that affect joining the military and remaining in military service are somewhat similar, although differences may exist due to political and regional differences among countries (Pákozdi and Bárdos 2022). This also indicates that the methods of analysis, as well as the management of the recruitment and joining process, may be significantly improved by techniques developed in other parts of the world—for example, Mosco's model (Taylor et al. 2015)—or by developing new skills (Cavaleiro et al. 2022), even in cases when external circumstances and political-economic situations differ. The results of the research presented in this paper show that enlistment motivations among the recruits in the Hungarian Defense Forces (Pákozdi and Bárdos 2022) can be maintained, and even improved, by the application of international experiences, while the model we developed provides a sound and appropriate basis for this process.

In summary, although our results are, in part, similar to many other studies, especially those related to Maslow's hierarchy of needs model, they also differ in many respects. What is truly innovative in the present study is the detailed description of the very complex needs and motivations structure of those serving in the Hungarian Defense Forces, as the highly effective grounded theory qualitative method yielded greater detail and generated a deeper understanding of the examined phenomenon. Based on our findings, we believe it is possible to initiate new processes and methods to improve both the subjective and objective characteristics of the Hungarian Defense Forces.

Another possible research topic (in relation to the military needs pyramid) is the investigation of the motivations of those serving in different categories (officers, non-commissioned officers, regular soldiers), with different legal relationships (contractual, professional), and of different genders and ages, which may result in a more advanced retention ability. Based on the star model, as well as on the new military needs pyramid, APPRECIATION emerges as an important research topic, since it appears to be a basic and general feature of military service motivation.

## 6. Limitations

The present study is not without limitations. As it was a qualitative study exploring military career needs, the findings cannot be generalized to broader populations of soldiers. The approach of the present study was exploratory, and it focused on obtaining a deeper understanding of participants' experiences.

During the study—due to the COVID-19 pandemic—it was difficult to identify and select working groups or military programs in which soldiers experienced the essentials of military life at different tactical levels (e.g., military exercises, NATO, EU, missions, trainings, etc.) during their service. The military programs were organized in line with the pandemic regulations: soldiers participated in the discussions in small numbers, which meant that several programs were needed to obtain the appropriate number of samples.

The study was facilitated by the fact that the lead author was involved as both a soldier and researcher. This generated trust among the participating colleagues, which we

regard as a positive feature of the research. When reading and analyzing the narratives, the lead author (...), as a military researcher, paid particular attention to specific linguistic expressions, such as "bidirectional weapon", "power", and "we are one team", which are rare in this form in everyday life and may, thus, mean different things to different people. Being a soldier means having a specialist's view of the research and being familiar with the military environment, which may have helped the work of the co-researcher.

However, this also meant that we had to be extra vigilant so as not to allow the lead researcher's military self to become too involved during the analysis. This was fortunately controlled by the "civilian eye" of the co-researcher, which helped turn this potential limitation into a positive contribution. We are grateful to the participants in the study, who contributed to the results with their trust and honesty.

Due to the COVID-19 pandemic and the state of emergency introduced by the Hungarian government to preserve the functionality of the country and the safety of its people, the soldiers of the Hungarian Army were performing tasks in hospitals and healthcare centers that did not require special qualifications (measuring temperatures, administrative and logistics tasks, providing directions, etc.) to ease the burden on health professionals. In addition, they were participating in border controls, which, combined with the pandemic-related work, imposed an additional psychological and physical burden on them. These factors contributed significantly to the results of the present research and to the better recognition of their motivations for remaining in the military.

In conclusion, we can state that, despite the initial difficulties, the results can be regarded as positive in that we have successfully broadened our knowledge of the applied methodology, as well as of military careers.

## 7. Publication Declaration

The authors declare that there are no conflicts of interest that could pose a real, possible, or apparent problem with respect to their participation in the publication. Furthermore, the lead author declares that, although she is a soldier and a member of the Hungarian Defense Forces, this has no impact on the findings and conclusions of the study. She also declares that the findings and conclusions obtained from the research are based exclusively on what was said by the respondents. The second author declares that he is not a member of the Hungarian Defense Forces and has no connections with the Hungarian Defense Forces that might have an impact on the findings and conclusions of the research.

**Author Contributions:** Conceptualization, M.P. (first author) and G.B. (second author).; methodology, M.P.; software, G.B.; validation, M.P. and G.B.; formal analysis, G.B.; investigation, M.P.; resources, M.P. and G.B.; data curation, G.B.; writing—original draft preparation, M.P. writing—review and editing, G.B.; visualization, M.P. and G.B.; supervision, G.B.; project administration, G.B.; no funding acquisition. All authors have read and agreed to the published version of the manuscript.

**Funding:** This research received no external funding.

**Institutional Review Board Statement:** The study was conducted in accordance with the Declaration of Helsinki, and the protocol was approved by the Ethics Committee of the ELTE Faculty of Education and Psychology. Project identification code is 2020/400, valid between 1 December 2020–1 March 2021.

**Informed Consent Statement:** All subjects gave their informed consent for inclusion before they participated in the study.

**Data Availability Statement:** Data are available and may be requested from the corresponding author.

**Conflicts of Interest:** The authors declare no conflict of interest.

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
