# Peer review of "A New Military Hierarchy of Needs Model"

_socsci, doi:10.3390/socsci11050217_

Round 1
Reviewer 1 Report
Being the topic maintaining a sounding interest in sociological research on the military in current times, it is a very "classic" subject, and this makes it a very difficult subject to be considered and studied in order to find out something new. With special reference to the sample, it's not a question of numbers of interviewees or of quantitative vs qualitative methods. Differences and similarities in comparison with previous literature should be considered and explained in a much deeper way. The new model proposed could be interesting but it's not adequately explained by means of research data, expecially in comparison with other mentioned models, more or less based on Maslow's theory, concepts should be adequately described in their meaning. And moreover, the choice not to consider data on gender, age, rank, time in service, and military branch (although explained) has turned into a serious difficulty to explain results.
Author Response
Dear Editor(s), Dear Reviewers:
Thank you for considering our article and giving an option to improve the manuscript. Below I describe the changes made.
First, both Reviewers asked for thorough language check. We sent the manuscript to an experienced native English speaker who has made a really complete check. You can see the changes in the manuscript in a different colour.
Of course, before this we had made several changes suggested by the Reviewers. The eliminated parts are now in blue, the added texts in green, this way you may follow all changes. Just for letting you to have a general impression, in another file we accepted all changes thus you may see the new version of the manuscript.
We have replaced the complete References part in the form required by Social Sciences, and added new resources, too.
There are two Figures which were suggested by the checking reader to be changed, but we couldn’t do that. Thus, the 2 new versions of Figure 2 and 3 are placed at the end of the manuscript and we ask to put them on their place.
Below we react (A) to the opinions of the Reviewers (R).
Reviewer 1
R1: Being the topic maintaining a sounding interest in sociological research on the military in current times, it is a very "classic" subject, and this makes it a very difficult subject to be considered and studied in order to find out something new. With special reference to the sample, it's not a question of numbers of interviewees or of quantitative vs qualitative methods. Differences and similarities in comparison with previous literature should be considered and explained in a much deeper way.
A1: Thank you for your opinion. We have added more references to compare our findings to those in the literature and made it with the older and newer publications. We strongly believe the qualitative analysis represents a different approach (see also below) and makes it possible to uncover hidden deeper factors.
R1: The new model proposed could be interesting, but it's not adequately explained by means of research data, especially in comparison with other mentioned models, more or less based on Maslow's theory, concepts should be adequately described in their meaning.
A1: Thank you for mentioning this. Yes, we in essence have found that Maslow’s theory represents a kind of reference even to new models or new approaches, anybody who wants to create a new model should refer to Maslow’s work and to deal with the differences. However, to make it clearer, we have added some new references which may help to compare our results to those obtained by other researchers.
R1: And moreover, the choice not to consider data on gender, age, rank, time in service, and military branch (although explained) has turned into a serious difficulty to explain results.
A1: Of course, in essence you are right. However, to make the responses of the interviewees honest and sincere, we omitted these factors to avoid their fear of being identified. In fact, based on our earlier experiences, they really mentioned several thoughts usually avoided during such interviews. We must add that we continue this study and definitely divide the (much larger) sample according to gender, age, rank etc.
Reviewer 2 Report
The work raises a very interesting issue concerning needs according to Maslow's hierarchy. This is an especially timely topic in the context of the sad war that is taking place in Ukraine. The topic of the article sounds very interesting, but I have the impression that it is unfinished, as if there is no further part to it, which, unfortunately, is reflected in the material that needs to be refined.
Unfortunately, in my opinion, the very weak side of the job is the lack of concrete figures. I believe that this aspect needs a major overhaul. I suggest that the authors rebuild both the codes and the star-shape model and complete them based on specific numerical data that will be the value of the work.
I propose to remove the aim of the paper nouns 16-32 to the end of abstract. There should be introduction first after then the aim of the paper. Authors write about GT analysis in 268 noun - vut there is lack more information about the background. I propose to expand the issue. Authors write that: " The interviews were run in 10-person groups..." (269-270 noun) lack information based on what guidelines were the respondents selected for the interview?
Another weak point of the study is the very poor literature review, which does not contain the most recent studies and is based only on old reports.
Author Response
Dear Editor(s), Dear Reviewers:
Thank you for considering our article and giving an option to improve the manuscript. Below I describe the changes made.
First, both Reviewers asked for thorough language check. We sent the manuscript to an experienced native English speaker who has made a really complete check. You can see the changes in the manuscript in a different colour.
Of course, before this we had made several changes suggested by the Reviewers. The eliminated parts are now in blue, the added texts in green, this way you may follow all changes. Just for letting you to have a general impression, in another file we accepted all changes thus you may see the new version of the manuscript.
We have replaced the complete References part in the form required by Social Sciences, and added new resources, too.
There are two Figures which were suggested by the checking reader to be changed, but we couldn’t do that. Thus, the 2 new versions of Figure 2 and 3 are placed at the end of the manuscript and we ask to put them on their place.
Below we react (A) to the opinions of the Reviewers (R).
Reviewer 2
R2: The work raises a very interesting issue concerning needs according to Maslow's hierarchy. This is an especially timely topic in the context of the sad war that is taking place in Ukraine. The topic of the article sounds very interesting, but I have the impression that it is unfinished, as if there is no further part to it, which, unfortunately, is reflected in the material that needs to be refined.
A2: Thank you for these remarks. First of all we have to make it clear that this study had been run before the war in Ukraine this is why it is not mentioned, although we agree with you regarding this fact.
We have added some new parts in which we summarize the new findings and draw more complete conclusions. Now we hope both the goal and the new results are clear.
R2: Unfortunately, in my opinion, the very weak side of the job is the lack of concrete figures. I believe that this aspect needs a major overhaul. I suggest that the authors rebuild both the codes and the star-shape model and complete them based on specific numerical data that will be the value of the work.
A2: The qualitative analysis, especially that applying the grounded theory (GT) method is characterized by understanding the words, sentences or paragraphs and by building a deeper and deeper and more and more complete expressions. To make it clear, we have added a detailed description of the process, which we cite below:
“During the axial coding process, our aim was to identify conceptual relationships among the open codes by investigating what a specific category can tell us about the basic problems and components and the different aspects of the studied phenomenon, and by looking for associations between individual codes and different dimensions. By the end of the axial coding process, basic categories and subcategories had been identified, allowing us to find relationships and correspondences. The notes taken in this phase contributed to the analysis in the next phase of the work. In the example shown in Figure 2, the 30 basic codes have been arranged into three groups under the representative axial codes. The three axial codes contain 12, 13, and 5 basic codes respectively, thus the number of basic codes collected into one axial code is not predetermined.
During the final, selective coding process, the categories and subcategories obtained in the first two phases were collected and grouped under new higher-level, summarizing codes and compared against the transcript (i.e., the correlations between the codes and the transcript were checked so that the correspondences could be validated). In this phase, we did not use the categories that had been omitted during the axial coding process. In the example, one selective code contains the three axial and the 30 basic codes (the so-called code tree).”
R2: I propose to remove the aim of the paper nouns 16-32 to the end of abstract. There should be introduction first after then the aim of the paper.
A2: Thank you for this suggestion with which we completely agree. This paragraph was replaced to the Methods section.
R2: Authors write about GT analysis in 268 noun - vut there is lack more information about the background. I propose to expand the issue.
A2: We added some more details to make this clear.
R2: Authors write that: " The interviews were run in 10-person groups..." (269-270 noun) lack information based on what guidelines were the respondents selected for the interview?
A2: We have added a detailed description of the process to make this clear.
R2: Another weak point of the study is the very poor literature review, which does not contain the most recent studies and is based only on old reports.
A2: Thank you for this important remark. Since we – on the first place – have concentrated to our results, missed to upgrade the literature background. Now, we had searched the literature for more resources and added several, relevant references.
We, again, thank both of you for the useful advices and hope now our manuscript has been significantly improved.
Sincerely yours,
Round 2
Reviewer 1 Report
Authors made a good job in revising their paper. Now it is much better, especially with regard to model definition and findings explanation.
Reviewer 2 Report
Thank you for improvement. Good job. I wish you all the best.